# Comparative Characterization of Different Cutting Strategies for the Sintered ZnO Electroceramics

**Jaka Dugar, Awais Ikram ***  **and Franci Pušavec ***

Faculty of Mechanical Engineering, University of Ljubljana, Aškerčeva Cesta 6, SI-1000 Ljubljana, Slovenia;
jaka.dugar@fs.uni-lj.si
\* Correspondence: rana.awaisikram@yahoo.com (A.I.); franci.pusavec@fs.uni-lj.si (F.P.)

**Abstract:** Sintered zinc oxide (ZnO) ceramic is a fragile and difficult-to-cut material, so finishing operations demand handling cautious and accurate surface tolerances by polishing, grinding, or machining. The conventional machining methods based on grinding and lapping offer limited productivity and high scalability; therefore, their incapacity to prepare tight tolerances usually end up with uncontrolled edge chipping and rough surfaces in the final products. This study investigates microstructural features with surface roughness in a comparative mode for conventional milling and abrasive waterjet cutting (AWJ). Edge topography and roughness maps are presented in this study to weigh the benefits of AWJ cutting over the conventional material removal methods by altering the feed rates. The porosity analysis implies that the differences during the multi-channel processing of varistors, which tend to alter the microstructure, should in turn exhibit a different response during cutting. The surface roughness, edge contours, and porosity generation due to shear forces are interpreted with the help of 3D optical and electron microscopy. The results demonstrate that the surface microstructure can have a noteworthy impact on the machining/cutting characteristics and functionality, and in addition, mechanical properties of ZnO varistors can fluctuate with non-uniform microstructures.

**Keywords:** ZnO; varistors; machining; milling; abrasive waterjet cutting (AWJ); porosity; electron microscopy; edge radius (r) profilometry; surface roughness ($S_a$, $S_{10Z}$); topography analysis



## 1. Introduction

Sintered ZnO is a class of functional material that is typically brittle with low fracture toughness, and final shaping requires careful handling and machining [1]. ZnO varistors are usually applicable as high surge current protective devices. Sintered varistors require smooth surfaces for the applying of metallization layers (Al, Ag paste) to join electrical leads/interconnects and contoured edges to adjoin the glass glaze along the sides [2]. Lapping widely used industrially in the finishing of inelastic ZnO ceramics and uncontrolled chipping or edge roughness in the finished products along with the high expense of machining costs and tool blunting necessitate alternative solutions [3,4]. Furthermore, the material removal rate (feed rates and turning velocity) are kept drastically low, which implies that current machining practices are poorly sustainable [1]. The conventional material removal methods imply large machining forces induced in the ZnO ceramic due to tool work piece contact, which results in vibration and chattering of the work piece. The highly brittle nature of ZnO ceramic exhibits a probabilistic tendency of fracture that leads to material damage and may also result in poor surface/edge finish. Moreover, due to chemical inertness and semiconducting behavior of ZnO, adopting suitable non-conventional machining is rather complicated. On the finished ZnO varistors, the roughened or shear force induced chipped edges are highly unfavorable for the functional properties; thus, care must be adopted in developing smooth surfaces and tapered edges.

Alternatively, the micro-machining approach can be favored over the conventional machining practices for developing ultra-precise ceramics with well-defined cutting geom-

etry (sharp and defined edges with radius r < 15 μm). Typically, the industrial waste of ZnO varistors failing during manufacturing stages, mechanical processing, or electrical impulse testing is estimated at approximately 15% of the total production. One way to enhance the machineability of the brittle materials is thermal activated machining of the ceramic so the heat is applied to entice localized softening in the areas to be machined [5–7].

Abrasive waterjet (AWJ) cutting represents an alternative approach to cutting very hard and brittle ceramics to good precision by the co-utilization of high-pressure water and (micro)abrasive particles for wet blasting of surfaces [8]. The added benefit of abrasive waterjet cutting apart from excellent tolerances also implies no inherent change in the surface microstructure due to lack of heat-affected zone (HAZ) formation [8–11]. It is conceivable that the AWJ method allows contoured edges, bevels, sharp corners, pierced holes, and profiles to be created with negligible inner radii [8,11,12]. The width of the cut (kerf) can be regulated by exchanging the parts within the nozzle, as well as varying the type and size of the abrasive. The kerf formed during nominal abrasive cutting lies in the range of 1.0–1.3 mm and can be made as narrow as 0.51 mm. AWJ cutting provides the capability of attaining accuracy down to 0.13 mm with repeatability up to 0.025 mm in subsequent passes [11,13,14]. The key factors including the AWJ cutting process include water jet pressure (available kinetic energy to cut), traverse rate (slower scan rate allows for higher volume fraction of abrasives on a given surface area that leads to higher surface finish and vice versa), abrasives (harder and larger sized particles support optimal cutting, necessary for ceramics and hard composites), abrasive flow rate (optimization of rapid material removal rate with respect to surface finish), standoff distance (between target and nozzle, defines kerf profile), and jet impingement angle (change in jet attack angle and directly influences target erosion/cutting) [15].

Hashish et al. [16] suggested that high pressure is more efficient in abrasive processes against the same power consumption. Jegaraj et al. [17] verified the effect of various AWJ parameters towards the machining responses so that the kerf width and depth of cut can be optimized by changing the feed parameters such that the surface quality does not vary significantly. Ma et al. [18] analyzed the kerf geometry with a light optical microscope and inferred that as the cutting speed increased, the kerf width subsequently decreased; thus, the kerf width increases for lower cutting speeds. Khan et al. [19] reasoned that by increasing the standoff distance, the waterjet widened, which also resulted in the broadening of the taper of the cut; however, high feed pressures reduced these effects. Shanmughasundaram et al. [20] argued that water pressure is a more significant parameter in AWJ than the transverse speeds and standoff distance to the workpiece. Srivastava et al. [21] reported the application of AWJ in lieu of shot peening for surface treatment of weldments.

Moreover, understanding the machining performance of ZnO varistors is imperative due to the prerequisite of an amorphous glassy coating along the sides as well as the metallization layer on the top and bottom surfaces; therefore, the geometric accuracy of these machined profiles is important for depositing dissimilar types of functional coatings on the same ceramic. Muženič et al. [1] reported on the machinability of ZnO varistors by laser-assisted milling (LAM) and suggested that machinability improved; the surface roughness and edge chipping reduction was realized by fine-tuning the laser power to 120 W. Except for this publication, the literature related to machining parameter optimization, even with conventional systems, and to the best of our research and knowledge is not available for ZnO sintered varistors. Consequently, this study investigates the abrasive waterjet machining performance of sintered ZnO varistors with respect to conventional milling to compare and obtain the parametric range (cutting mechanism), causing edge chipping or transgranular/sudden failure in hard varistor ceramics. Predominantly the edge chipping occurs due to Bi-rich spacer phase coerced out, which in turn causes high surge current catastrophically flowing through the edge of the varistors as the easier (less resistance) path rather than through the ZnO grains [22,23].

Important parameters classified and compared within this study include the surface roughness and contour mapping, topographical imaging, and porosity count (areal

fraction), between the AWJ cutting and conventional milling. The surface integrity and roughness characterization made thru the optical roughness measurement system and the scanning electron microscopy infer directly that the machined surface roughness is interlinked with the occurrence of grain pull-out during the milling and cutting operation. The porosity distribution analysis provides insight into the effect of shear forces during cutting by interlinking the difference in sintered densification with the effects causing edge chipping/roughness and grain pull-out (ZnO as well as secondary phases).

Further, this work correspondingly elucidates why the conventional machining (milling) is not appropriate due to edge chipping in ZnO varistors owing to their low fracture toughness of 2.16 MPa·m$^{0.5}$, which worsens in relatively poorly dense sintered ceramics, particularly along the edge regions.

## 2. Experimental Methodology

The initial value of the edge radius for the pristine non-machined ZnO samples could be averaged from ~110 to 125 μm (surface roughness parameters (arithmetical mean height—$S_a$ = 1.37 μm and ten-point height—$S_{10Z}$ = 25–40 μm), as shown in Figure 1 with the geometric specifications of the samples. The ZnO varistors' geometric specifications are illustrated in Figure 1, having a nominal diameter of 42 mm, thickness of 12.6 mm, and edge radius r ≤ 130 μm, and the ten-point surface roughness $S_{10Z}$ was categorized in a range from 25 to 40 μm.

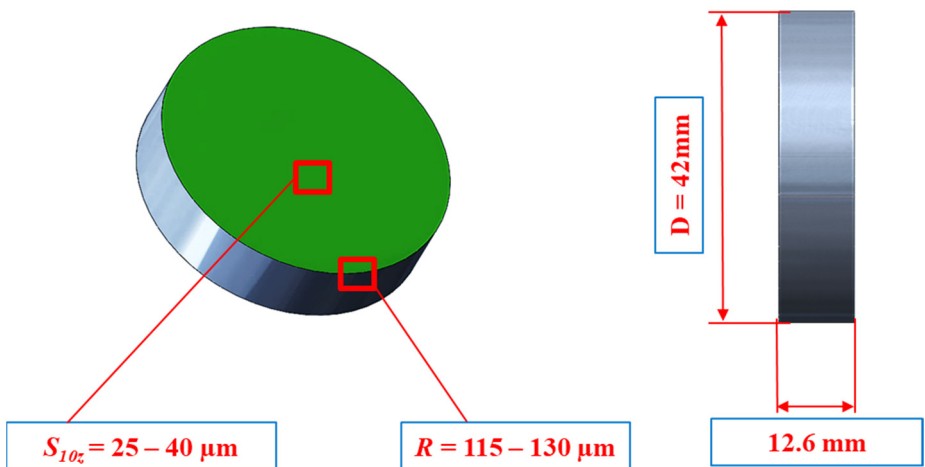

**Figure 1.** Geometry of ZnO varistor used in milling and AWJ cutting operations.

The ZnO ceramics were commercially produced at Varsi d.o.o Slovenia and were developed with the fixed lower-plunger uniaxial pressing of green compacts prior to the sintering operation above 1200 °C, and thus the two sides could have different milling and surface roughness features; hence, the color markings are presented as shown in Figure 2. The relative green density was assumed to be >60% after uniaxial pressing, whereas the sintered samples had relative density above 95%, on which further milling and AWJ cutting operations were performed.

The top side marked with red color shows that this section directly received the compaction pressures during uniaxial pressing, which specifies a relatively higher green density on the top (red) part, whereas the blue (bottom) part designates the section away from the pressure plungers. These descriptive color markings of red and blue sides are repeatedly used in the results and discussion part.

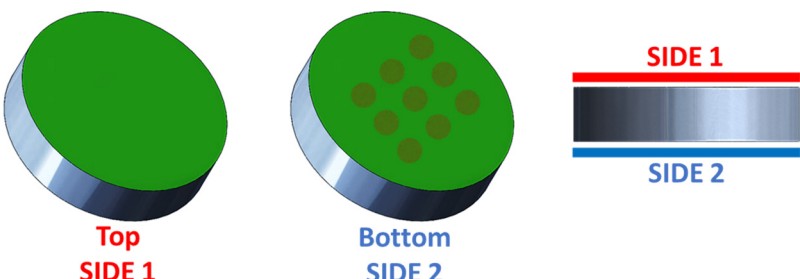

**Figure 2.** The division of ZnO ceramics turning operations to two sides, illustrative of uniaxial pressing of green compacts prior to sintering operation.

Conventional milling was performed using a CNC Mori Seiki SL 153 milling machine with a polycrystalline diamond (PCD) type cutting tool (C-13581) having a single edge as shown in Figure 3. The following milling parameters were employed to machine the surfaces of the ZnO varistors: axial depth of cut ($a_P$) = 0.1 mm and width of cut ($a_e$) in relation to the diameter of the cutter = 5 mm; feed velocity of 400 mm/min; and spindle speed (n) of 4000 rpm/min ($V_C$ = 100 m/min and $f_Z$ = 0.04 mm). All the milling operations were carried out in dry cutting conditions. All the ZnO to be milled samples were mounted and fixed to the same tightness on the CNC unit.

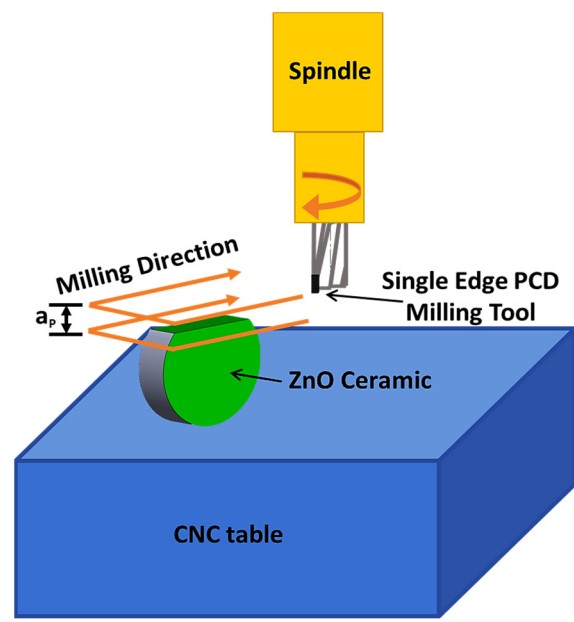

**Figure 3.** Schematics of milling operation on ZnO varistor.

The abrasive waterjet cutting was performed with the Bohler ECOTRON 403 system equipped with a 2652A JetMachining Center high pressure water pump, and Garnet mesh-80 abrasive (Jetstar International STAN/80/1000). The schematic illustration of the abrasive waterjet machining setup utilized in this study is presented in Figure 4. Cutting parameters with AWJ include feed rate range of 120–1060 m/min, garnet abrasive flow rate of 0.45 kg/min, standoff distance (nozzle height to specimen) *f* = 2 mm, focused/focusing nozzle diameter of 0.8 mm, water nozzle diameter of 0.3 mm, and water pressure at 300 MPa (3000 bars). The feed velocities (*v*) 120 mm/min and 160 mm/min yielded the same results, which why throughout the text, the slowest feed rate option is 160 mm/min.

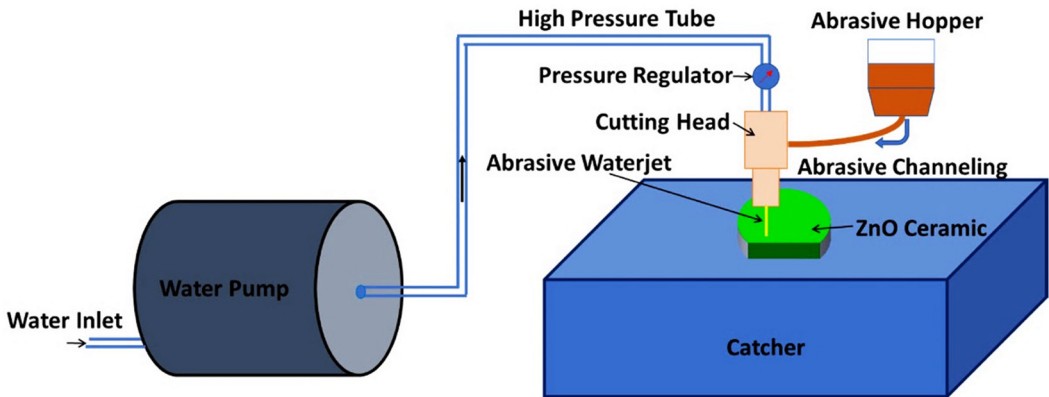

**Figure 4.** Schematic illustration of waterjet machining setup.

The waterjet cut ZnO specimens in their cross-section as shown in Figure 5 were grinded by 500 and 1000 grit SiC papers on rotary units and then polished with 500 nm diamond paste residue on the velvet cloth to prepare them for microscopical examination.

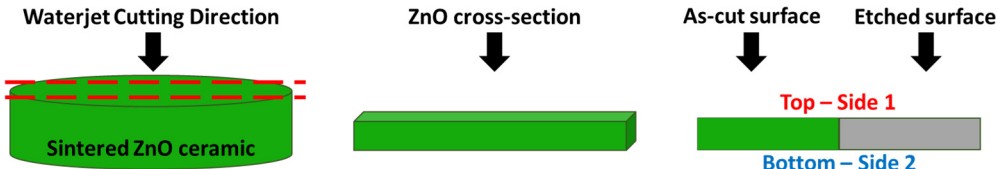

**Figure 5.** Abrasive waterjet cutting operation on ZnO varistors.

The assessment of surface roughness and contour mapping was made with an Alicona InfiniteFocusSL measurement system in the form of 3D scans of the surface and edges, generated in two passes. The sintered ZnO workpiece was attached in a tilted position within the specimen holder and top-down optical light illumination. Using the shallow depth of field in the optical system and by vertically scanning (piezoelectric positioning system across the *z*-axis), the topographic data in the form of a color scheme by varying the focal length was obtained on randomly selected 5 mm long profiles in the *x*-axis direction. Minimum measurable profile roughness—$R_a$ at 10X and 20X magnification was 0.3 μm and 0.15 μm, respectively. A repetition of surface roughness measurements was considered along the *y*-axis, approximately 10 μm apart, to obtain the mean values.

The microscopic imaging was made with a JEOL 7600F field emission scanning electron microscope at an accelerating voltage of 20 kV in secondary electron (SE) mode to obtain the surface and topographical view of the ceramics, and an in-focus backscattered electron (BSE) detector was employed to acquire the phase contrast data.

## 3. Results and Discussion

Metal oxide varistors (MOVs) are voltage-dependent resistors (VDRs) having nonlinear non-ohmic current–voltage features that offer safety against high surge current [23]. The non-linear characteristics depend strongly on the microstructure, which is developed intrinsically for a certain chemical composition and applied processing methods [24]. The microstructure of a typical MOV comprises approximately 90% of dark grey ZnO matrix grains a few tens of microns in size, as shown in Figure 6. For a nonlinear non-ohmic response, the varistor forming oxide (VFO), mainly involving $Bi_2O_3$, is added to the composition. To augment the threshold voltage and surge current endurance, the number of intergranular (IG) layers in series and parallel, respectively, need to be maximized. Hence, for high surge shielding, finer matrix ZnO grains and high grain boundary (GB) surface area are prerequisites [22,25–29]. In addition to VFO, $Sb_2O_3$, $Al_2O_3$, $Cr_2O_3$, $SnO_2$,

$MnO_2/Mn_3O_4$, NiO, and $Co_3O_4$ are added in minor amounts for gaining non-linearity factor ($\alpha$) enhancement, grain size refinement, Schottky barrier height elevation, and inversion boundary (IB) generation [22,23,27–29] for high transient voltage protection [26].

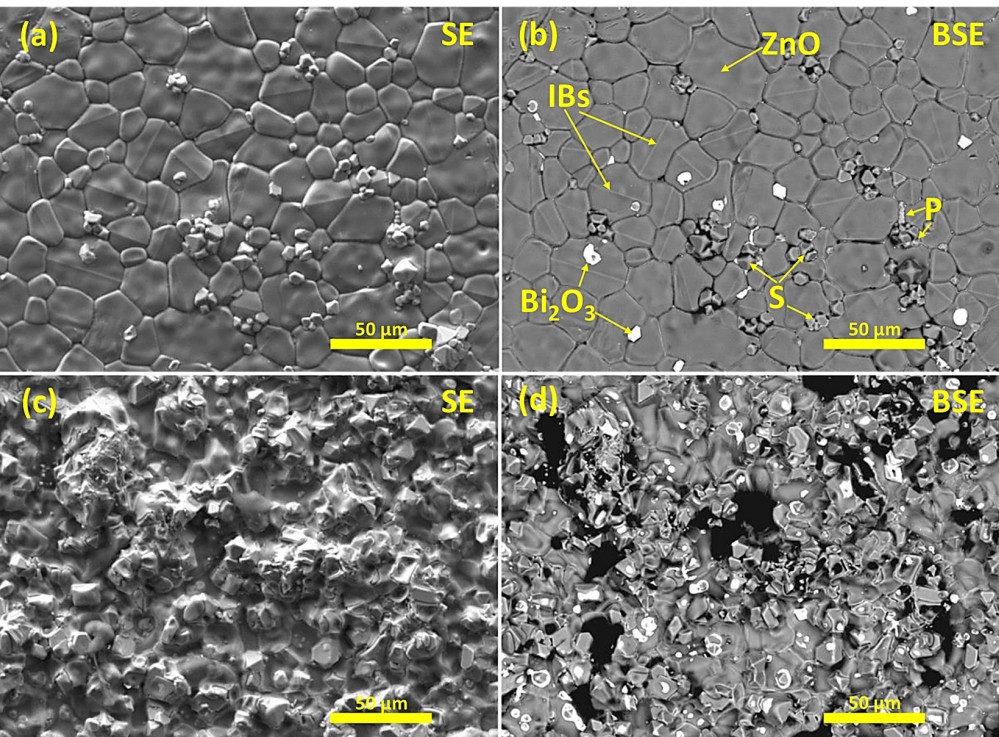

**Figure 6.** SEM images of ZnO varistors after sintering with (**a**,**b**) having backing ceramic (refractory $Al_2O_3$ crucible) to cover the bottom surfaces, whereas (**c**,**d**) shows sintering of ZnO varistor green compact freely in air, i.e., top side. Here (**a**,**c**) represent secondary electron (SE) imaging for topographical features, while (**b**,**d**) represent phase contrast and microstructural composition in backscattered electron (BSE) imaging mode.

The sintering conditions can have a significant impact on the end microstructure of ZnO MOVs, as shown in Figure 6. The green samples were placed on the refractory ceramic boat during the sintering operation, so the bottom surface was in contact with the refractory, whereas the top surface remained in contact with air. With the ceramic $Al_2O_3$ crucible backing layer present during the sintering, the end products are usually flat, and the microstructure is well defined, as shown in Figure 6a,b. These alumina crucibles (>99% $Al_2O_3$) are routinely used for the sintering operations up to 1700 °C and may contain traces of silica ($SiO_2$) and magnesia (MgO). Thus, the bottom part usually retains a well-defined and smooth microstructure, which is unlike the case for the top side, which is in contact with air, as shown in Figure 6c,d.

The topography indication (secondary electron imaging—SE) is shown in Figure 6a, and well aligned flat ZnO grains were present with very little porosity. The phase contrast image in Figure 6b with back scattered electron (BSE) detector clearly illustrated the presence of a $Bi_2O_3$ intergranular phase (IP). The $\alpha$-$Bi_2O_3$-rich IP at the grain boundaries (GBs), triple pockets, and ZnO grain junctions appeared as bright regions in the microstructure, of widths from 0.1 to 1 μm, reliant on sintering conditions [22,26]. The $Sb_2O_3$ oxide was supplemented to restrict the ZnO grain growth and boost the solubility above the 740 °C eutectic of Zn in the IG phase. The $Sb_2O_3$ oxide supports the phase transformation above 900 °C to the spinel ($Zn_7Sb_2O_{12}$) phase (S) shown in Figure 6b along the IG, and this reaction is activated by pyrochlore ($Bi_3Zn_3Sb_3O_{14}$) phase (P) decomposition above this temperature range during sintering [27,28]. The spinel (S) phase appears in light greyish tone enclosed within the bright IG phase, whereas the pyrochlore (P) phase can exist as

scattered whitish precipitates at the GBs. For Sb/Bi ratios < 1 in the composition, the liquid phase appeared due to eutectic melting (ZnO–Bi$_2$O$_3$) at 740 °C, while for Sb/Bi ratios > 1, the IG phase adhered to the P phase, and the S-rich liquid phase formed after higher temperature decomposition. The P-phase decomposition to the S-phase and the IG phase created a high degree of non-linearity characteristics in MOVs. Undoubtedly, without Sb$_2$O$_3$, the Schottky barrier height was reduced, which upsurged the donor density of charges, causing leakage current increase and lessening of non-linearity characteristics. These dopants intuitively decreased the mobility of ZnO-Bi$_2$O$_3$ GBs due to the pinning effect of the spinel grains, therefore augmenting the surge shielding [22,23,25,28,29].

The free sintering in air caused a very rough microstructure, as shown in the SE image in Figure 6c. By observing this free air sintering in BSE mode, Figure 6d confirms that the secondary phases were scarce and homogeneously distributed in the non-flat ZnO matrix exposed to air. This variation in microstructure due to sintering most certainly may cause different machining characteristics. Therefore, a thorough analysis on material removal/cutting performance changes in the two sides, given that green compaction behavior was different, was carried out in this work.

The qualitative difference of edge surface profiles after milling and AWJ cutting is shown in Figure 7. Conventional milling developed quite rough edges, as can be seen in Figure 7a and at higher magnification in Figure 7b, whereas the abrasive waterjet (AWJ) cutting yielded smoother surface finishes at the circumferential segments of the varistor, as seen in Figure 7c,d. The average S$_a$ and S$_{10Z}$ values for the milling experiment yielded 3.69 and 103 μm, respectively. Further characterization was enabled by an Alicona 3D measurement system for variations in the surface topography, roughness, and slice edge features comprehensively based on different feed rates of AWJ cutting and most optimal milling results only. This generic comparison can help to devise a cutting strategy for avoiding a qualitatively poorer surface finish at the edges, which is not acceptable for the deposition of glass glaze.

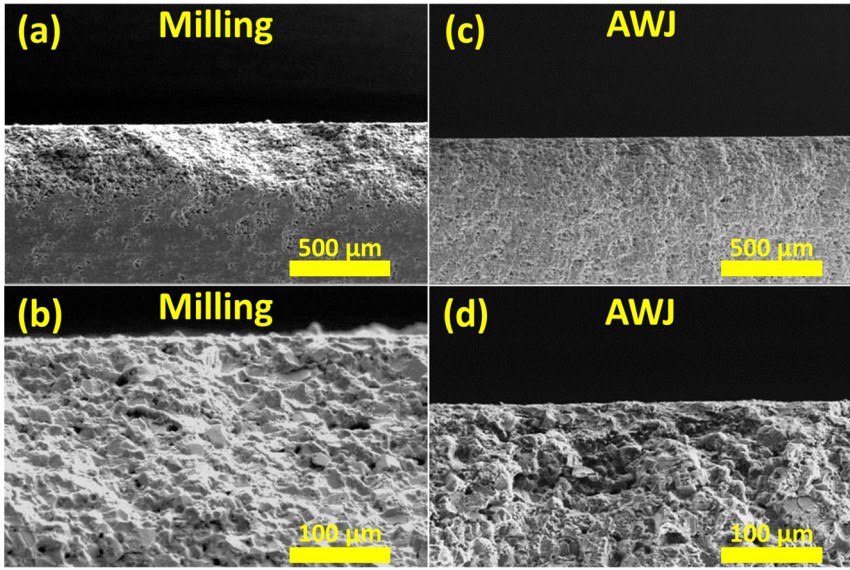

**Figure 7.** The edge machining difference in SE imaging mode of (**a**) conventional milling and (**b**) at higher magnification of depression area, and (**c**) optimal abrasive waterjet (AWJ) cutting at *v* = 160 mm/min, and (**d**) smooth AWJ cross-section magnified to 1000×.

A comparative edge profile assessment was also carried out following the SEM analysis of the milled and waterjet cut samples, as illustrated in Figure 8 based on variations in height (z-axis) per cross-sectional width. The edge radius of the ZnO sintered varistor prior to machining persisted in the range of 100–125 μm, as shown in Figure 1. With controlled milling, the edge radius was reduced to a size range 83.5 μm, determined with

the Alicona system and presented in Figure 8a. The waterjet cut samples provided two extremities of operation between the slowest feed rate of 160 mm/min and the most rapid at 1060 mm/min. At the most rapid conditions, shown in Figure 8b, the edge trim was even worse than the optimally milled sample, with radius values exceeding 164.5 μm on average. However, at feed rate $v$ = 160 mm/min, the resultant slice roughness at the edges was trimmed down to 65 μm. This implies unsuitability of AWJ cutting at higher feed rates, as the edge regions and taper did not adhere to the principal electrical impulse testing requirements [30].

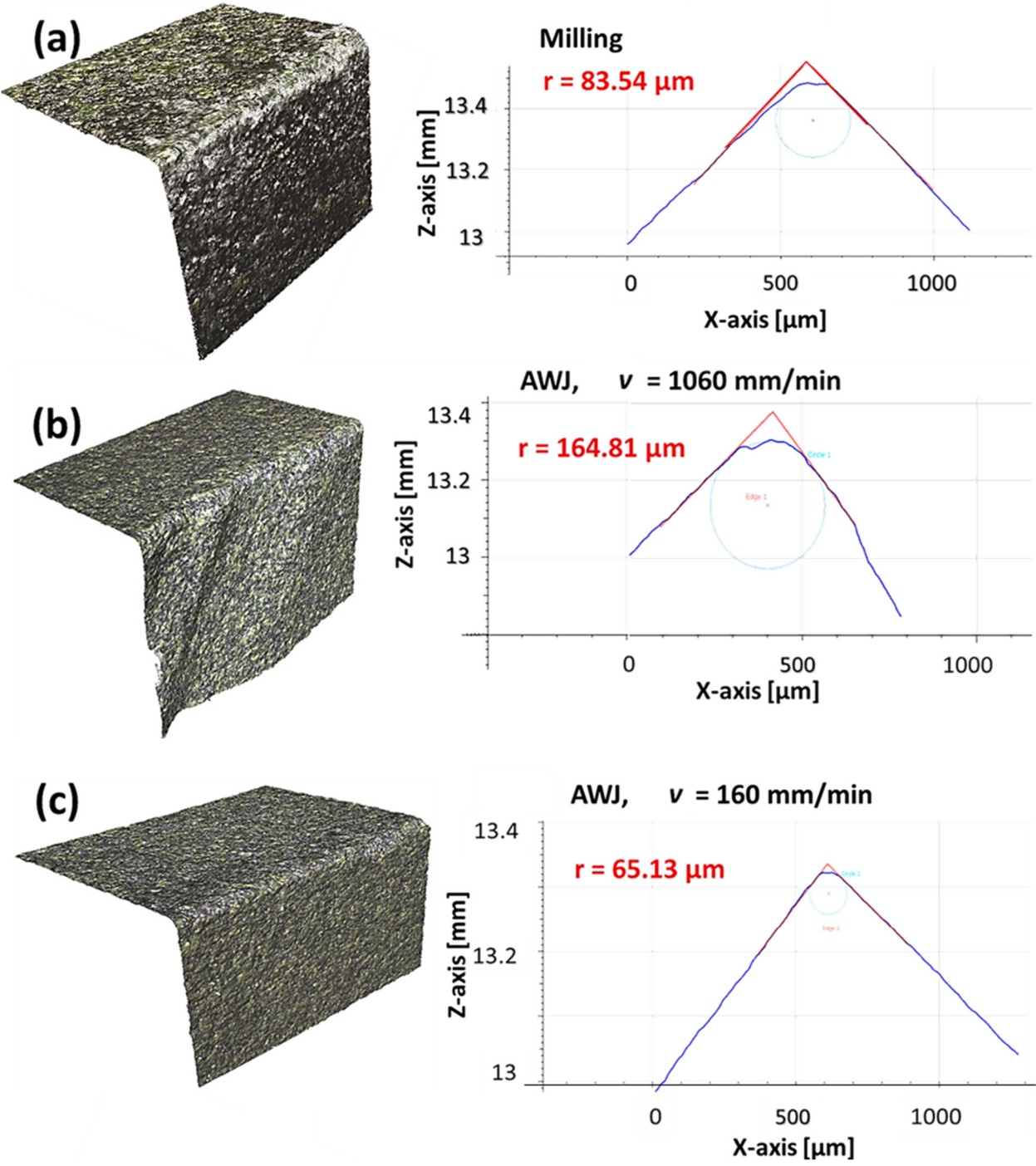

**Figure 8.** The edge profile comparison between the (**a**) optimally milled and (**b**) the coarsest edge with feed rate 1060 m/min, and (**c**) the best AWJ cut ceramic at the feed $v$ = 160 mm/min.

Further on, the comparison of surface roughness was also made by investigating the topographical color scans. The surface roughness after the optimally controlled milling experiment resulted in the cross-section shown in Figure 9. The color-coded regions had $S_a$ values of spot (1) of side 1—the top side at 3.42 μm, region (2) at 3.35 μm, the (3) central region of cross-section at 3.44 μm, and the average of complete cross-section in (4) at 3.54 μm after optimum milling operation. The 3D color contrast height variation (topographical) map suggests the central regions in the cross-section retained height profiles usually under 10–15 μm with peak–valley contours after milling. However, the edge profiles were consistently coarser beyond the −15 μm roughness range.

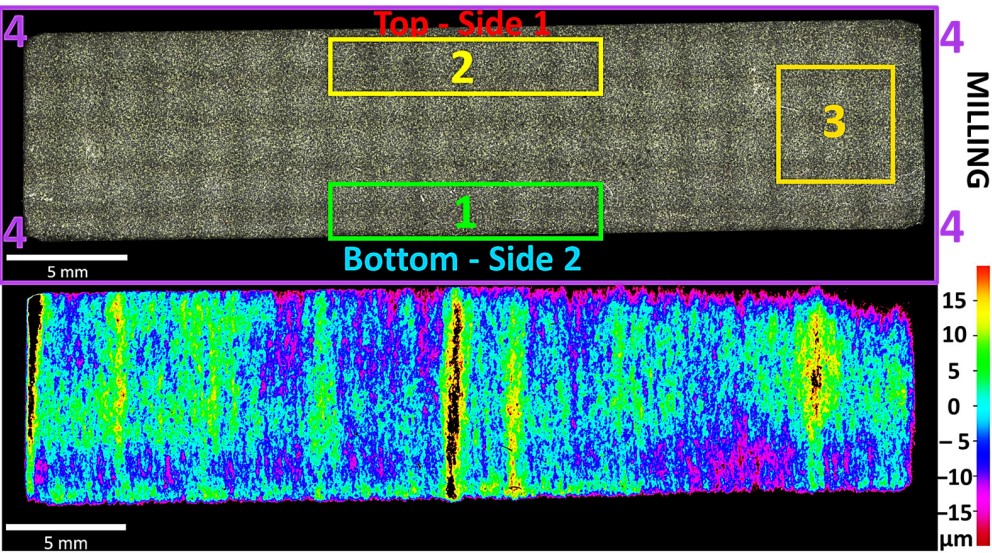

**Figure 9.** Qualitative analysis of roughness profile and topographic mapping for height variation after controlled milling.

The variation in the topographical features (valleys–heights) of the cross-sectional slices that delineated the surface roughness by factors $S_a$ and $S_{10Z}$ were evaluated by the Alicona 3D system for all the AWJ cutting feed rates (*v*), as can be seen in Figure 10, to develop a comparison with the controlled milling approach. The Sa values increased by 74% at higher feed rates of 1060 mm/min as compared to the *v* = 160 mm/min. The trend was clear, and with an increase in the feed rate, the resultant $S_a$ and $S_{10Z}$ values increased in the cross-sections sliced by AWJ.

It is also worth noting that the lower parts of the samples usually indicate higher color deviation with this effect amplifying at higher feed rate, evidently representing larger variation in surface roughness values. This happens due to the jet expansion effect at higher feed velocities, such that the jet interacts with the surface initially at the top side of the specimen in Figure 10. At lower feed rates (up to 500 mm/min), the jet remained essentially linear, so the top and bottom cut cross-sections had similar $S_a$ and $S_{10Z}$ values. However, the cross-sections made with 895 and 1060 mm/min feed rates had much coarser bottom sections due to jet expansion and cutting action occurring in wider areas than linear channeling at slower feed velocities. The average Sa and S10Z values are reported in Figure 10 for each respective feed rate. In view of damage reduction, it is essential for the settings utilized in the current AWJ cutting setup to retain controlled low feed rates, delivering diminished roughness in cut slices.

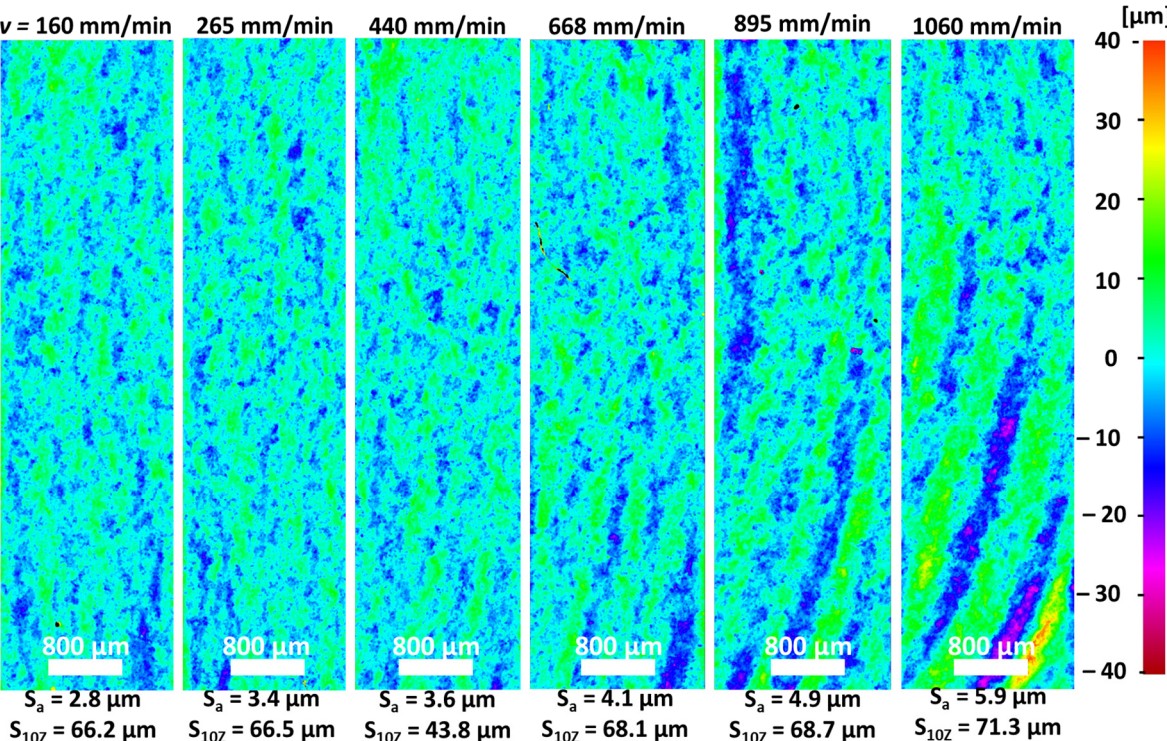

**Figure 10.** The variation in surface roughness parameters for various feed rates (*v*) in AWJ cutting.

In contrast, the surface roughness after AWJ cutting of cross-section of specimen with feed rate (*v*) = 160 mm/min can be seen in Figure 11 with the marked regions having $S_a$ values of (1) in the bottom, side 2, at 2.84 μm, spot (2) for the side 1 at 2.89 μm, the central region (3) of cross-section at 2.88 μm, and the average of the whole specimen cross-section marked by (4) at 2.82 μm. The 3D color contrast height variation map indicates excellent surface finish at the central regions of the cross-section by AWJ cutting, but sparsely minor edge tapering beyond the 10–15 μm roughness range. Nonetheless, Figures 7 and 8 confirmed that the edge roughness in conventional milling was high compared to AWJ cutting, and consequently the $S_{10Z}$ value for AWJ cutting corresponded to 87 μm, which is a degree lower than the conventional milling experimental result at 103 μm.

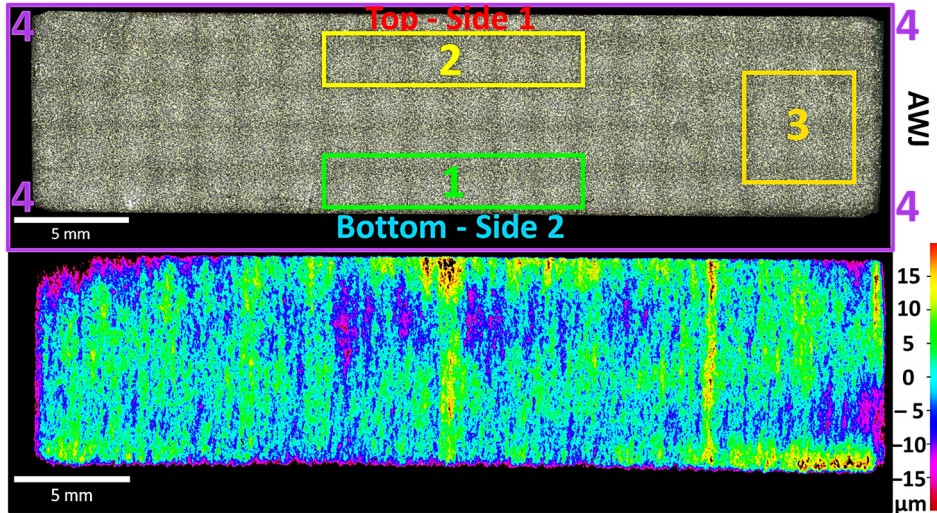

**Figure 11.** Measurement of roughness profile and topographic mapping for height variation in the cross-section after AWJ cutting for sample cut at *v* = 160 mm/min.

The porosity analysis was performed with MATLAB Binary Segmentation to evaluate the chipping and material removal mechanism under the applied shear loading in machining/cutting. Figures 12 and 13 presents two sides after AWJ cutting and microstructure variation due to processing parameters.

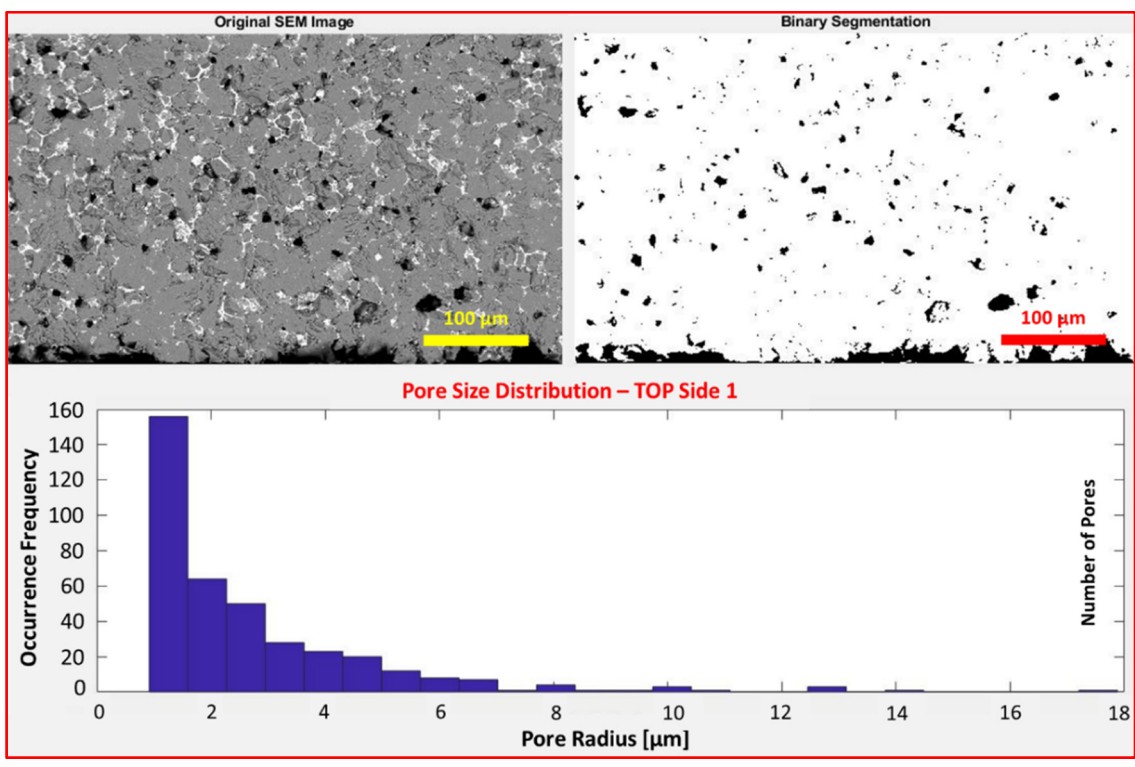

**Figure 12.** Quantitative porosity analysis on the top, side 1, of the cross-section after AWJ cutting.

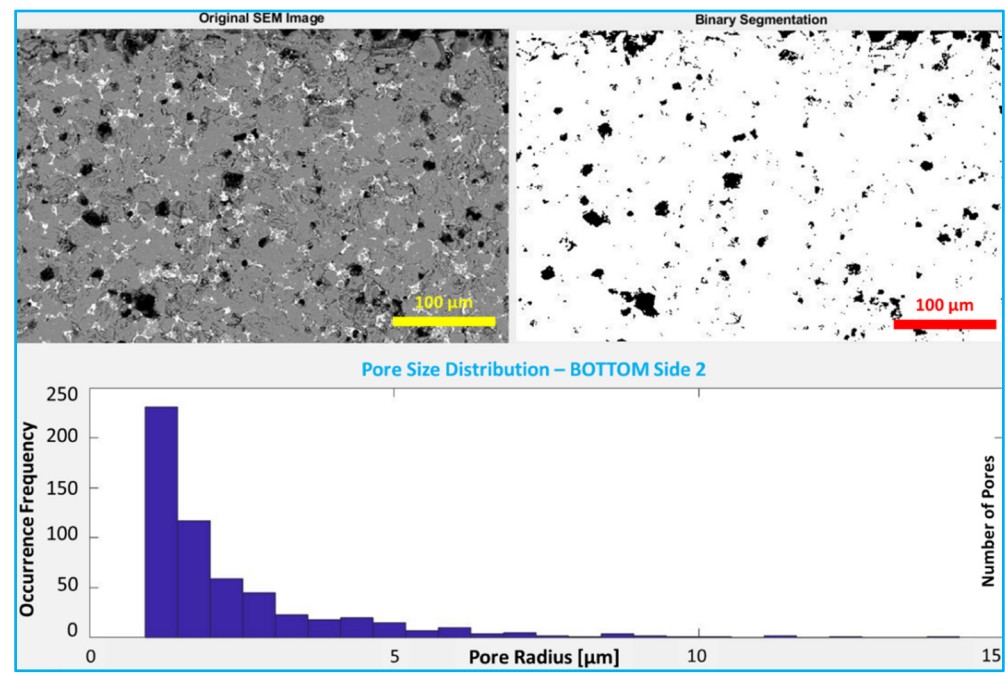

**Figure 13.** Porosity distribution analysis on the bottom, side 2, of the cross-section after AWJ cutting at *v* = 160 mm/min.

Figure 12 shows the quantitative porosity distribution in the top, side 1, of the AWJ cut cross-section, indicating high number of pores in the range of 1–2 μm size within the microstructure. Very large pores above 7 μm were sparse, which indicates these pores were present within a uniform matrix of normal sized ZnO grains in the range of 5–15 μm. A more uniform microstructure retaining edge taper and minor edge chipping was observed, as few larger pores were distributed close to the edge. For sintered ZnO ceramics under large, applied shear forces during the conventional machining, the edge chipping was expected to be higher in the side where the microstructure was more uniform. Likewise, the grain pull out was further perceptible in the region of the microstructure where the distribution of secondary phases was higher with the ZnO matrix. Figure 12 implies that this top side experienced higher pressure during the compaction stage, and later with normal ZnO grain growth, the present pores were eliminated to a large extent. However, applied shear forces during machining and AWJ cutting caused larger brittle ZnO grains expulsion at the edges, whereas small secondary phases were pulled out from the middle of the cross-section.

In Figure 13, it can be seen that a similar porosity distribution in the range of 1–3 μm for the bottom, side 2, was approximately two-fold higher than the top, side 1. Not only were the larger pores present within the bulk of the microstructure, but although minor, the edges were also not as uniform as the top side. The pores' areal distribution for the bottom, side 2, elucidated high edge chipping and internal grain pull-out in less dense parts of the sintered ceramic. This suggests that the variation in green compaction pressures and processing conditions had a significant influence on the microstructural integrity and resulting mechanical properties. Typically, with lower green density, the sintering can cause particle coalescence only to an extent such that internal porosity still remains widely distributed [31,32]. In this case, it is more apparent from Figure 13 that larger size and regular grain shaped pores had formed, which suggests that ZnO grains were pulled out of the matrix instead of only the smaller secondary phases in Figure 12 within a denser matrix. Highly non-uniform cross-sections and edges effectively rendered these functional ceramics useless for high transient or surge protection applications, as the surface metallization and circumferential insulation were non-uniformly deposited on the varistor.

Large lateral shear loads produce very rough edges, which suggests AWJ cutting as a more appropriate surface peening and circumference tapering tool than conventional milling and machining. Non-uniform edges present persistent difficulty in applying dielectric epoxy or glass layers to shield the varistor during assemblage. Moreover, high surface roughness in center faces causes poor metallization deposition, resulting in surge protectors typically failing impulse tests in quality assessment rather than going onward with real-world utility [26]. The AWJ cutting with feed rates 160 mm/min and up to 440 mm/min yielded very similar $S_a$ values and the edge radius profiles in Figure 8. However, the feed rate above 500 mm/min and up to 1060 mm/min from Figure 10 prove that the obtained $S_a$ and $S_{10Z}$ values are far worse than controlled milling. On the contrary, the average surface roughness ($S_a$) maintained by AWJ cutting at 2.82 μm is exceptionally better than conventional machining counterparts >3.69 μm, and thus this report justifies their industrial potential prior to surface and edge preparation. The AWJ cutting mechanism is quite complex, involving a series of compressive, tensile, shear forces leading to abrasion, friction, erosion, wear, and cracking. Characteristically, the compressive stresses higher than the compressive fracture strength applied by waterjet media lead to generation of a kerf (which depends on jet pressure, traverse speeds, standoff distance, and the nozzle diameter). The threshold pressure to create the kerf is associated with the principal compressive stresses ($\sigma_C$) by a factor of 0.2 in plain waterjet cutting. However, this model cannot be simply applied to AWJ cutting, since this factor becomes quite complicated with the presence of abrasive particles, and multiple models still do not present values for threshold pressure in the brittle materials. Nevertheless, experimentally it has been proven that the depth of cut and penetration increases with higher abrasive flow rate. Typically, the depth of cut is inversely proportional to the traverse rate, as in

the case of continuous standoff distance variation, due to widening of the solid jet and higher interaction of the disintegration zone particles with the brittle surface, leading to a rougher profile. In this study, the standoff distance was already optimized to 3 mm, and feed rates were tweaked. The wear zone shown in the lower section of the samples cut by AWJ at higher feed in Figure 10 represents the effect of higher tangential shear and abrasion assisted erosion, e.g., in case of 1060 mm/min sample, the $S_a$ values are much higher (6 μm) than the controlled milling setup at 3.6 μm. Similarly, the edge profile shown in Figure 8b represents this impact of high feed rates, leading to pronounced tapering of rough edges >165 μm and broader shear zones.

Further analysis is underway to link the binary segmentation, chip evolution, and mechanical properties with the variations in jet pressure, stand-off distance, traverse speed, and different jetting media on the final roughness characteristics of ZnO varistors. Increasing the standoff distance causes waterjet widening, which also results in broadening [19], such as that observed for higher velocity cut samples, causing a rougher surface. Increasing the water pressure may circumvent these issues [20]; however, such parameters and their impact on brittle ZnO electroceramics remain a subject of successive investigations.

## 4. Conclusions

In this comparative study on the microstructural features after conventional milling and abrasive waterjet cutting, we report exceptionally less surface roughness from the latter technique. Optimization of the cutting/machining parameters is closely associated with the microstructure, suggesting that the edge chipping is dominant in the denser side where the microstructure is more uniform, while the grain pull-out mechanism (both ZnO and secondary phases) becomes obvious in the region where more secondary phases are distributed in/along the matrix. The binary segmentation analysis of porosity implies that the densely sintered side has lower terminal porosity and finer edges after AWJ cutting, whereas the other side retained a higher number of pores, apparently due to the larger grain pull-out and pronounced ZnO grain chipping. The larger shear forces during conventional machining cause ZnO ceramics to become unusable for high transient or surge protection utility, since the surface metallization and edge insulation will be heterogeneously coated on the varistor, so AWJ cutting is a much better alternative for the surface preparation requirements.

**Author Contributions:** Conceptualization, F.P. and A.I.; methodology, A.I. and J.D.; software, A.I. and J.D.; validation, A.I.; formal analysis, J.D.; investigation, J.D. and A.I.; resources, F.P. and J.D.; data curation, J.D. and A.I.; writing—original draft preparation, A.I.; writing—review and editing, A.I. and J.D.; visualization, A.I. and J.D.; supervision, F.P.; project administration, F.P.; funding acquisition, F.P. All authors have read and agreed to the published version of the manuscript.

**Funding:** The authors acknowledge the financial support from the Slovenian Research Agency performed under the program code P2-0266, and the project serial numbers L2-8184 and L2-1836.

**Institutional Review Board Statement:** Not applicable.

**Informed Consent Statement:** Not applicable.

**Data Availability Statement:** Not applicable.

**Acknowledgments:** We would also like to thank Slavko Bernik from the Department of Nanostructured Materials at the Jožef Stefan Institute Slovenia for his support in performing the scanning electron microscopy analysis on the machined ZnO ceramics. Thanks also to Damir Grguraš for the help with additional proofreading of the manuscript.

**Conflicts of Interest:** The authors declare no conflict of interest.

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
