# Peer review of "Comparative Characterization of Different Cutting Strategies for the Sintered ZnO Electroceramics"

_applsci, doi:10.3390/app11209410_

Round 1
Reviewer 1 Report
The submitted manuscript discusses the effect of two types of cutting techniques on the microstructure of ZnO varistors. Whilst the observations have been clearly described, the manuscript lacks the depth of analysis required to establish a comprehensive comparative study of the two cutting methods. In particular, the in-depth understanding of cutting parameters in both techniques is missing. For example, the authors could have investigated the effects of variables such as jet pressure, stand-off distance and traverse speed, on final roughness characteristics of the components. Furthermore, the authors have not clarified whether the shear load regimes present in the two techniques were comparable. This need to be clarified and backed up by numerical calculations. The authors claim that “parameter optimization is closely associated with the microstructure” however no evidence of the optimization process is presented, as far as I can establish only one set of parameters were used for cutting. Given the above shortcomings, I do not view this manuscript to be suitable for publication at its current format.
Author Response
Please find attached revision report.
Thank you and kind regards

Reviewer 2 Report
The manuscript deals with experimental investigations on the cutting of sintered ZnO ceramics with different methods. The results are of interest and practical value. The following suggestions for improvement are made prior to publication:
(1) It should be better emphasised by the authors that new scientific knowledge has been gained with the investigations. Instead, the anticipation of results at the end of section 1 "Introduction" can be omitted.
(2) Providing documented comparative data on the improvement in processing time and processing quality by abrasive waterjet cutting compared to conventional methods would enhance the significance of the manuscript.
(3) Since only one setting was chosen by the authors for each of the processing methods investigated, an outlook on further reasonable investigations could be given in the summary.
(4) In lines 134 ff. "a_e" and "a_p" are probably mixed up.
(5) Can the proportions of the additives be provided? (line 179).
(6) The "backing ceramic" described in the figure 6 caption and in the text, lines 190 ff. is not entirely clear and should be briefly explained.
(7) In figure 8 the signs of the legend should be visible.
(8) When naming authors in the text, the first author should be named first (e.g. lines 74 and 77) and in the bibliography the source reference to [14] should be checked for completeness.
Author Response

(The authors gave the same response as above.)

Reviewer 3 Report
The “Comparative characterization of different cutting strategies for the sintered ZnO electroceramics” manuscript reports the microstructural features of ZnO ceramics micro-machined by using conventional milling (CM) and abrasive waterjet (AWJ) techniques. The authors present results on the surface roughness, edge contours and porosity of the machined samples. These technical results are well organized and presented in a good manner. The paper is interesting and may deserve publication in Applied Sciences journal even though some point should be addressed:
1. Abstract, from line 19 to line 21
The authors claim that “The results demonstrate that green and sintered density can have a noteworthy impact on the machining/cutting characteristics and the functionality as well as mechanical proper ties of ZnO varistors can fluctuate with non-uniform microstructure”. However, ZnO samples with various green density and/or bulk density cannot be identified in the “Results and discussion” section.
2, Experimental Methodology, from line 116 to line 117
The geometric specifications are provided for as-sintered ZnO samples?
3. Experimental Methodology, from line 120 to line 123
All the fabrication steps should be detailed!
4. Experimental Methodology, from line 129 to line 130
The authors claim that the samples exhibit “a relatively higher green density on the top (red) part”. Did they measure the green density of ZnO samples?
5. Results and Discussion, line 195
What do the authors mean by "sintering conditions"? Did they vary the parameters of the sintering process (e.g. sintering temperature, sintering time)?
6. Results and Discussion
It will be great to have in Fig. 7 SEM images with higher magnification for both ZnO samples.
7. Results and Discussion
The size of the font used in Fig. 9 and Fig. 10 is too small.
8. Conclusions, from line 305 to line 307
The authors claim that “In this comparative study on the microstructural features after conventional milling and abrasive waterjet cutting, we report exceptional less surface roughness from the latter technique.”. However, the reported surface roughness of the micro-machined ZnO samples is 2.9 for CM and 2.82 for AWJ. I would not call this an exceptional difference!
9. Conclusions, from line 307 to line 310
The authors concluded that “The cutting/machining parameters optimization is closely associated with the microstructure suggesting that the edge chipping is dominant in the denser side where microstructure is more uniform, and grain pull-out (both ZnO and secondary phases) becomes obvious in region where more secondary phases are distributed in the matrix.”. However, the authors the authors do not measure bulk density of the edges and the conclusion is rather speculative.
10. There are several typing error.
Author Response

(The authors gave the same response as above.)

Round 2
Reviewer 1 Report
The manuscript has improved significantly both in terms of the depth of analysis as well as clarification of the suitability of the experimental design in addressing the research hypothesis. Whislt the lack of FEM numerical analysis is clear the supporting discussion provided in the revised manuscript explains the challenges around the analysis and provides the reader with a more comprehensive perspective on the complexity of experimental evaluations in AWJ processes. I recommend acceptance of the article in the current format.
Author Response
We are very glad to read the judgement of the reviewer following their suggestions to modify the experiments as well as results section in the previous amendment of the manuscript. We do hope to correlate the complex FEM modelling with our experimental AWJ cutting data, once all the output parameters have been clearly identified and interlinked. Lastly, we thank the reviewer again for their worthy comments, some critical suggestions in terms of scientific clarity and soundness of the message to the readers as a prerequisite to publishing with MDPI Applied Sciences. These helpful remarks have to be credited for improvement in the quality and legibility of the manuscript.
In case you may find additional segments to correct or comments/advice, we would be obliged to facilitate you without hesitation.
Reviewer 3 Report
The revised version of the applsci-1347511-peer-review-v2 manuscript entitled “Comparative characterization of different cutting strategies for the sintered ZnO electroceramics” have been significantly improved. I suggest a minor revision to strengthen the paper. I provide the following suggestions for this purpose:
1. Abstract, line 16
I suggest the authors to revise the “porosity analysis verifies” expression. No direct porosity measurements were presented in this manuscript.
2. According to the “Instructions for Authors”, the “Acronyms/ Abbreviations/ Initialisms should be defined the first time they appear in each of three sections: the abstract; the main text; the first figure or table. When defined for the first time, the acronym/abbreviation/initialism should be added in parentheses after the written-out form.”. The authors should check the manuscript again for proper use of abbreviations (e.g. line 52, line 115, line 380).
3. Results and Discussion, line 254
Please check the accuracy of the scale bars in Figure 7.b and Figure 7.d
4. Results and Discussion, line 255
Delete “shows” word from figure 7 caption.
5. References must be rewritten in accordance with the “Instructions for Authors”.
Author Response
Please find answers and changes within the attachment.
Kind Regards
